# Systematic review of exercise for the treatment of pediatric metabolic dysfunction-associated steatotic liver disease

Martha R. Smith[1,2], Elizabeth L. Yu[1,3], Ghattas J. Malki[1], Kimberly P. Newton[1,3], Nidhi P. Goyal[1,3], Karen M. Heskett[4], Jeffrey B. Schwimmer[1,3]*

1 Department of Pediatrics, Division of Gastroenterology, Hepatology, and Nutrition, University of California San Diego School of Medicine, La Jolla, California, United States of America, 2 Chicago Medical School, Rosalind Franklin University of Medicine and Science, North Chicago, Illinois, United States of America, 3 Department of Gastroenterology, Rady Children's Hospital San Diego, San Diego, California, United States of America, 4 The Library, University of California San Diego, La Jolla, California, United States of America

* jschwimmer@ucsd.edu

**Data Availability Statement:** All relevant data are within the manuscript and its Supporting Information files.

## Abstract

### Background & aims

Steatotic liver disease affects approximately 1 in 10 children in the U.S. and increases the risk of cirrhosis, diabetes, and cardiovascular disease. Lifestyle modification centered on increased physical activity and dietary improvement is the primary management approach. However, significant gaps in the literature hinder the establishment of exercise as a targeted therapeutic strategy for pediatric metabolic dysfunction-associated steatotic liver disease (MASLD), previously known as nonalcoholic fatty liver disease (NAFLD). We performed a systematic review of studies assessing the impact of exercise interventions on validated hepatic outcomes in children with NAFLD.

### Methods

We searched CENTRAL, PubMed, Embase, Web of Science, CINAHL, and Google Scholar on June 5 and 6, 2023, for studies in English involving children aged 0 to 19 years diagnosed with NAFLD or at increased risk for NAFLD due to overweight or obesity. We updated the search on August 8, 2024. Eligible studies were required to examine the impact of exercise interventions on hepatic steatosis or liver chemistry. The risk of bias was assessed with RoB2 and ROBINS-I. Data extraction was performed by two independent reviewers.

### Results

After screening 1578 unique records, 16 studies involving 998 children were included. This comprised seven studies comparing exercise intervention with non-exercising controls, three uncontrolled studies of exercise intervention, two studies comparing exercise plus lifestyle interventions with lifestyle interventions alone, and nine studies comparing different types of exercise interventions. Five of the 11 studies that evaluated hepatic steatosis

**Funding:** The author(s) received no specific funding for this work.

**Competing interests:** The authors have declared that no competing interests exist.

reported an absolute decrease of 1% to 3%. In the nine studies that evaluated liver chemistry, no significant changes were observed.

## Conclusions

Evidence supporting exercise intervention for the treatment of pediatric MASLD is limited. Existing studies were constrained by their methodological approaches; thus, there is a pressing need for high-quality future research. This will enable the development of precise, evidence-based exercise guidelines crucial for the effective clinical management of this condition.

## Introduction

Steatotic liver disease in the form of nonalcoholic fatty liver disease (NAFLD) is the most prevalent chronic liver disease among children aged 2 to 19 years, with approximately 1 in every 10 children exhibiting hepatic steatosis [1, 2]. This prevalence increases significantly in children with obesity, affecting approximately 1 in 4 children within this group [3]. Males are more commonly affected than females, and prevalence is higher among adolescents compared to younger children. Rates also vary by race and ethnicity, with the highest prevalence reported in children from South America and Asia [4]. To better reflect its systemic implications, NAFLD has been renamed metabolic dysfunction-associated steatotic liver disease (MASLD). MASLD encompasses a broad spectrum of disease severity, including isolated steatosis in its mildest form, that can progress to metabolic dysfunction-associated steatohepatitis and lead to more severe hepatic complications of cirrhosis and hepatocellular carcinoma. MASLD can also be associated with extrahepatic manifestations such as cardiovascular disease and type 2 diabetes mellitus [5]. Thus, the development of evidence-based interventions for MASLD is crucial.

In the absence of specific pharmacological interventions, lifestyle modification, primarily focusing on dietary improvements and increased physical activity, represents the cornerstone of MASLD management [2]. Although studies suggest that lifestyle modifications targeting weight loss can moderately reduce hepatic fat and alanine aminotransferase (ALT) levels in affected children, a critical void exists in understanding the specific impact of exercise as a therapeutic strategy [6–8]. Notably, existing research predominantly combines exercise interventions with broader lifestyle modifications, hindering the elucidation of exercise's independent effects on MASLD. Furthermore, the existing research varies greatly in the type, frequency, intensity, and duration of exercise studied, making it difficult to develop standardized exercise recommendations for MASLD patients [9]. Given that increased physical activity is a key component in the primary treatment of pediatric MASLD, developing specific, evidence-based exercise guidelines is essential for enhancing the clinical management of this condition.

Our primary aim was to investigate the effect of exercise on liver disease in pediatric patients with NAFLD and/or MASLD. Given the evolving terminology, we retained the use of "NAFLD" throughout this review for consistency, as most of the included studies were published before the nomenclature change. We conducted a comprehensive systematic review to examine the impact of exercise interventions on liver health in children diagnosed with NAFLD or at heightened risk due to overweight or obesity (BMI $\geq 85^{th}$ percentile). This review aims to synthesize existing literature to better understand exercise's role in managing steatotic liver disease in children and to identify gaps that warrant further investigation.

## Methods

We conducted a systematic review according to the Cochrane guidelines for systematic reviews of interventions [10]. We adhered to the Preferred Reporting Items for Systematic reviews and Meta-Analyses (PRISMA) guidelines in our synthesis [11]. The protocol is in the supporting document titled S1 File. We reviewed studies that investigated the effects of exercise intervention on validated hepatic outcome measures in children with NAFLD or at higher risk for NAFLD due to overweight or obesity.

### Types of studies

We included full-text English language publications of interventional studies, including randomized controlled trials (RCTs), non-randomized controlled trials (non-RCTs), and uncontrolled trials published in peer-reviewed journals. Studies were only eligible for inclusion if they reported on outcomes relevant to our review objectives.

### Types of participants

We included studies that evaluated children and adolescents (aged 19 years or younger) diagnosed with NAFLD or considered at heightened risk due to overweight or obesity, defined as a BMI $\geq 85^{th}$ percentile. This criterion aligns with clinical guidelines for screening children at risk for steatotic liver disease [2]. Including these populations ensures a more comprehensive understanding of exercise's potential impact across the full spectrum of pediatric patients at risk for NAFLD and/or MASLD. Participants aged 19 years or younger were selected in accordance with the World Health Organization's classification of children and adolescents, which includes individuals up to 19 years of age. This approach aligns with established practices in pediatric gastroenterology and reflects the age range commonly managed in pediatric gastroenterology clinics in the United States. Notably, the renaming of NAFLD to MASLD did not affect the eligibility criteria of the included studies, as the terminology change reflects a conceptual shift rather than a change in the core diagnosis, and pediatric guidelines have not yet made any modifications based on this shift.

### Types of interventions

We included studies that evaluated the following comparisons: exercise versus no exercise; exercise plus lifestyle intervention versus lifestyle intervention alone; one type of exercise versus another type of exercise; and one type of exercise plus lifestyle intervention versus another type of exercise plus lifestyle intervention. We also considered uncontrolled studies that implemented exercise-only interventions. Exercise sessions could take place in any setting, including supervised or self-directed environments, and could be conducted individually or in groups. All types of exercise-based interventions were eligible for inclusion, such as aerobic training, resistance training, or a combination of both, regardless of frequency, intensity, or duration. Co-interventions like nutrition counseling, psychological counseling, or motivational interviewing were permitted, provided they were equally administered across comparison groups.

### Types of outcome measure

We evaluated three primary outcome categories: quantitative imaging assessment of hepatic steatosis, liver chemistries, and liver histology. These measures were chosen because they are the most clinically relevant, widely accepted for clinical trials, and recommended in clinical practice guidelines for MASLD.

Quantitative imaging methods, such as magnetic resonance imaging (MRI) and magnetic resonance spectroscopy (MRS), were prioritized because they provide precise, non-invasive assessment of liver fat content. MRI is validated against histology, showing high sensitivity and specificity in detecting and quantifying hepatic steatosis [12–18]. As such, MRI is a reliable tool for monitoring therapeutic responses over time. In contrast, conventional ultrasound was excluded due to its limited sensitivity, operator dependence, and inability to quantify liver fat accurately, making it unsuitable for assessing treatment effects [19, 20].

Liver chemistries, specifically ALT and gamma-glutamyl transferase (GGT), are widely used as biomarkers of liver injury and dysfunction. Changes in these enzymes have been shown to strongly correlate with improvements or worsening of liver histology in children with NAFLD, including steatosis and inflammation [21]. Thus, they serve as valuable, non-invasive indicators of liver health and therapeutic response in pediatric populations.

Histological assessment remains the clinical reference standard for diagnosing MASLD and evaluating liver pathology, including steatosis, inflammation, and fibrosis [2]. While biopsies are invasive and less often performed in pediatric populations, histological outcomes were considered when available to provide the most comprehensive analysis of liver changes. The inclusion of these measures ensures that our review reflects the most robust, clinically meaningful data available on the impact of exercise interventions in children with MASLD.

## Search methods for identification of studies

**Electronic searches.** Our electronic database search was conducted on June 5 and June 6, 2023, covering the following databases: Cochrane Central Register of Controlled Trials (CENTRAL) in the Cochrane Library (inception to Issue 6, 2023), PubMed (PubMed.gov) (inception to 5 June 2023), Embase (Embase.org) (inception to 5 June 2023), Web of Science Core Collection (inception to 6 June 2023), and CINAHL Complete on EBSCOhost (inception to 5 June 2023). We restricted the search to English publications, with no limitations on the date or location of studies. A search specialist was consulted to ensure the comprehensiveness of the search strategy, which is detailed in the S2 File. As part of the peer review process, we re-ran the search on August 8, 2024, incorporating the new terms "MASLD" and "metabolic dysfunction-associated steatotic liver disease" to align with the evolving terminology for steatotic liver disease.

**Searching other resources.** In addition to database searches, we examined reference lists from all primary studies and relevant reviews to identify any interventional studies that may have been missed. Additionally, on June 5, 2023, we performed a targeted keyword search on Google Scholar (www.scholar.google.com), collecting the first 300 search results. For the updated search on August 8, 2024, we applied a date filter from 2023 to 2024 and collected the first 100 results (S2 File).

## Data collection and synthesis

**Selection of studies.** The search results were imported into the web-based software Rayyan, and duplicates were systematically eliminated. Two review authors (MS and EY) independently assessed the titles and abstracts for potential inclusion, resolving any discrepancies through discussion and arbitration by a third review author (JS) where necessary. Following the initial screening phase, we obtained full-text articles of potentially eligible trials and applied the same procedures as for titles and abstracts to assess their suitability for inclusion against our predefined criteria. Detailed reasons for the exclusion of full-text articles were recorded. The selection process was meticulously recorded and presented in a PRISMA flow diagram.

**Data extraction and management.** We developed and piloted a standardized data extraction sheet, adapted from the Cochrane Data Extraction and Collection Form, to ensure reliable

and accurate extraction of pertinent data [22]. Two review authors (MS and EY) independently extracted study, participant, and outcome data. Discrepancies were resolved through discussion, with consultation from a third review author (JS) when necessary.

We extracted several categories of data from each study. General information included the study title, study ID, author(s), year of publication, journal or source, and country of origin. Study characteristics encompassed the diseases studied, study design, study date, duration of participation, and setting. Participant characteristics involved a description of the study population, inclusion and exclusion criteria, method of recruitment, sample size, use of clusters (if applicable), method of randomization, baseline imbalances between arms, withdraws and exclusions, age rage, mean age, sex, race/ethnicity, height, weight, BMI, liver fat, ALT, aspartate aminotransferase (AST), GGT, diagnoses, and comorbidities. Intervention details captured the number of participants randomized to each group, a detailed description of the exercise type, frequency, intensity, duration of each session, duration of the treatment period, co-interventions, integrity of delivery (whether prescribed or supervised), and compliance. Outcome information included data on both primary and secondary outcomes, time points measured, assessment methods, and statistical methods used. Finally, we collected any additional information, including key conclusions reported by the study authors.

Following data extraction, we organized and synthesized the information according to study design and outcomes. For each study, we provided a detailed summary and reported the absolute mean difference for each identified outcome. We also created tables summarizing baseline participant characteristics, exercise protocols, and study results. Data were presented as reported, without imputation or adjustment for missing values. Given the limited number of studies on this topic, no subgroup analyses were planned or performed. Additionally, due to the limited data available for each outcome, the high heterogeneity across studies in terms of design, population, intervention, and outcome measures, and a high risk of bias in 10 out of 16 included studies, a meta-analysis was not performed [23].

**Assessment of risk of bias in included studies.** Two review authors (MS and EY) conducted independent evaluations of each study using either the Cochrane risk-of-bias tool for randomized trials (RoB 2) or the Risk Of Bias In Non-randomized Studies of Interventions (ROBINS-I) tool, as outlined in the *Cochrane Handbook for Systematic Reviews of Interventions* [10]. In cases of discrepancies, a third reviewer (GM) conducted an independent evaluation of the study. Final decisions were reached through discussion among the full author team to ensure consistency across all assessments. All judgments were made in accordance with guidance from the *Cochrane Handbook for Systematic Reviews of Interventions* [10]. Details are provided in S3 File. The findings of our risk of bias assessment were visually summarized using the web application robvis [24].

**Comparison studies to provide clinical characteristics of pediatric patients with NAFLD.** In order to assess the generalizability of the studies included in our review to the clinical pediatric NAFLD population, we examined clinical trials involving children with biopsy-proven NAFLD. We created tables to summarize and compare the baseline characteristics of children with biopsy-proven NAFLD to those depicted in the studies encompassed by our review. We examined participants' sex, mean age, mean levels of ALT, AST, and GGT, and liver fat measures. In cases where baseline values were provided for each intervention group, we utilized a weighted average technique to calculate an overall mean for the study population.

## Results

### Results of the search

A comprehensive search was conducted in June 2023 and updated in August 2024, yielding a total of 1578 records from our database queries. Supplementary searches, including Google

Scholar and citation chaining, contributed an additional 1079 records. After removing duplicates, 1714 unique records remained. Through title and abstract screening, we narrowed down the selection to 26 full-text articles. Following a rigorous assessment, we identified 16 reports from 16 distinct studies that fulfilled the prespecified inclusion criteria. Of these 16 studies, one was identified through the updated search conducted in August 2024 [25]. There were 10 full-text articles excluded for diverse reasons, as detailed in Fig 1. A visual representation of our search process is provided through a PRISMA flow diagram (Fig 1).

Our review encompasses 16 studies, including 11 randomized controlled trials, two non-randomized controlled trials, and three uncontrolled trials. Pertinent participant characteristics have been synthesized for reference in Table 1, while a comprehensive breakdown of protocols and outcomes can be found in Table 2. Hepatic fat was evaluated using quantitative liver imaging methods in 11 studies, while ALT or GGT changes were assessed in nine studies. Our search did not identify any studies using liver histology. Detailed intervention descriptions can be found in S4 File.

## Summary of risk of bias assessment

We assessed risk of bias using the RoB 2 tool for randomized controlled trials and the ROBINS-I tool for non-randomized and uncontrolled studies of interventions. The results of the risk of bias assessments are shown in Figs 2 and 3. Among the 11 randomized control trials included in this review, seven studies were rated with a high concern for bias, two with some concern, and two with low concern. The domains with the most significant risk of bias were deviations from intended intervention and bias in outcome measurement. The high risk of bias occurring due to deviations from the intended intervention largely occurred due to participant non-adherence to the assigned intervention regimen without an appropriate analysis to estimate the effects of adhering to the intervention [28, 31, 36]. Bias due to deviations from the intended intervention also occurred due to issues with the trial context that led to participants being excluded from the analysis, with a potentially substantial impact on the results [25]. We discuss the high risk of bias in outcome measurement in the discussion section, "Use of validated assessment tools." Among the two non-randomized and three uncontrolled studies, one was determined to have a critical concern for bias, two had a serious concern, and two had a moderate concern. Confounding was the largest contributor to potential bias in the non-randomized and uncontrolled studies.

## Exercise only interventions

Our search yielded 10 studies that investigated the effects of exercise interventions alone on hepatic lipid content and liver chemistries in children with or at risk for NAFLD due to overweight or obesity. These studies include seven controlled trials and three uncontrolled trials.

**Studies of exercise interventions compared with a control group.** Lee, S. et al. conducted a RCT in 45 adolescent males with obesity. The primary aim of the study was to evaluate the effect of aerobic versus resistance exercise on abdominal adiposity, ectopic fat, and insulin sensitivity [33]. Intrahepatic lipid was measured by MRS in a subset of participants (n = 29). The exercise groups participated in three 60-minute sessions per week over a 12-week period. Participants were not required to have hepatic steatosis to participate. Moreover, liver chemistry was not measured. At baseline, the mean liver fat fraction was normal in all three groups, thus most participants did not have hepatic steatosis. The baseline mean liver fat fraction ranged from 2.2 to 4.7%. Following the 12-week intervention, the aerobic exercise group demonstrated a 1.0% absolute reduction in hepatic fat (4.7% to 3.7%), and the resistance exercise group demonstrated a 1.1% absolute reduction in hepatic fat (2.9% to 1.8%).

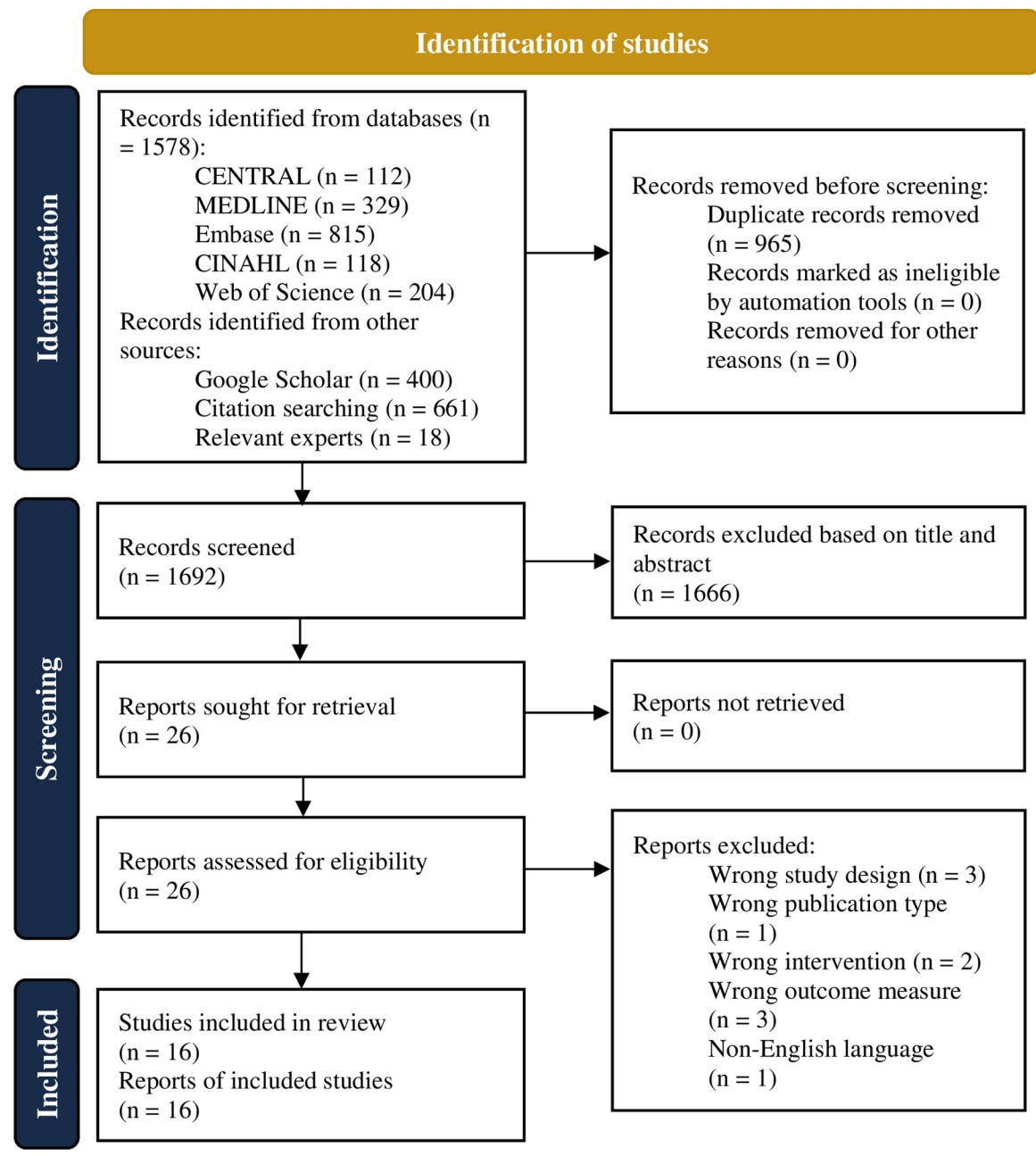

**Fig 1. PRISMA flow diagram.**

Lee, S. et al. conducted a randomized trial in 44 adolescent girls ages 12 to 18 years with obesity to evaluate the effect of aerobic versus resistance exercise on intrahepatic lipid content measured by MRS [34]. The exercise groups participated in three 60-minute sessions per week over a 12-week period. Participants were not required to have hepatic steatosis to participate. Consequently, at baseline, most participants did not have hepatic steatosis; the mean hepatic fat fraction ranged from 2.0 to 3.0%. After the intervention, the aerobic group exhibited a 0.95% absolute reduction in liver fat, while no change was observed in the resistance group.

**Table 1. Participant characteristics of included studies.**

| Study | Study sample size | Age, years | Female sex, % | ALT, U/L | AST, U/L | GGT, U/L | Liver fat measure, % |
|---|---|---|---|---|---|---|---|
| Bell 2007 [26] | 14 | 12.7 | 43 | 26 | — | — | — |
| Van der Heijden 2010(a) [27] | 15 | 15.6 | 53 | 39 | — | — | 9[a,b] |
| Lee, YH. 2010 [28] | 54 | — | 20 | 32 | 26 | — | — |
| Van der Heijden 2010(b) [29] | 12 | 15.5 | 50 | — | — | — | 9.2 (2.9)[a] |
| Davis 2011 [30] | 38 | 15.8 | 100 | — | — | — | 6.8 (3.6)[c] |
| Hasson 2012 [31] | 100 | 15.4 | 61 | — | — | — | 6.1 (2.2)[c] |
| De Piano 2012 [32] | 58 | 16.5 | 53 | 30 | 24 | 21 | — |
| Lee, S. 2012 [33] | 45 | 14.9 | 0 | — | — | — | 3.3 (2.9)[a] |
| Lee, S. 2013 [34] | 44 | 14.8 | 100 | — | — | — | 2.4 (3.5)[a] |
| Hay 2016 [35] | 106 | 15.2 | 24 | — | — | — | 6.0 (6.2)[a] |
| De Lira 2017 [36] | 84 | 14.8 | 75 | 26 | 20 | — | — |
| Lee, S. 2019 [37] | 118 | 14.4 | 64 | — | — | — | 2.6 (3.6)[a] |
| Labayen 2020 [38] | 116 | 10.6 | 53 | — | — | 16[d] | 5.4 (3.7)[c,e] |
| Iraji 2021 [39] | 34 | 13.1 | 0 | 38 | 34 | — | — |
| González-Ruíz 2022 [40] | 120 | 13.4 | 68 | 18 | 21 | 19 | 225 (42) dB/m[f] |
| Tas 2023 [25] | 37 | 15.2 | 57 | 22 | 22 | — | 5.9 (3.9)[g] |

Hepatic steatosis grade was not measured in any study. Abbreviations: ALT, alanine transaminase; AST, aspartate aminotransferase; GGT, gamma-glutamyltransferase; —, no data available.

[a]Liver fat assessed by MRS.

[b]Value approximated from a bar graph. No numerical data was provided.

[c]Hepatic fat fraction assessed by MRI.

[d]GGT values were available for 97 participants who completed the intervention.

[e]Hepatic fat fraction was measured for 115 participants.

[f]Hepatic steatosis assessed by FibroScan with controlled attenuation parameter.

[g]Intrahepatic triglyceride content assessed by MRI-proton density fat fraction (MRI-PDFF).

Iraji et al. performed a study of 34 male adolescents aged 10 to 15 years with suspected NAFLD. All participants had liver ultrasonography suggestive of hepatic steatosis and a one-time measure of ALT that was above the biological upper limit of normal [39]. Participants were randomly assigned to one of three groups for 8 weeks: high-intensity interval training (HIIT), a school-based exercise program, or control. Exercise sessions were three times per week for either 36–40 minutes (HIIT group) or 50–60 minutes (school-based group). The primary study outcome was change in ALT. The baseline ALT values were mildly elevated. The study found a decrease in ALT of 4 U/L for both the HIIT group and the school-based exercise group.

Davis et al. conducted a RCT in 38 female adolescents with obesity to assess the effect of exercise on adiposity and insulin sensitivity [30]. They evaluated the effect of the intervention on MRI hepatic fat fraction. Participants were not required to have hepatic steatosis in order to participate. Liver chemistry was not evaluated. Participants were randomized into three distinct groups: exercise, exercise supplemented with motivational interviewing, and control. The exercise interventions consisted of 60–90-minute supervised sessions of combined cardiovascular and strength exercises conducted biweekly for 16 weeks. At baseline, the mean hepatic fat fraction in the intervention groups was slightly elevated and did not change with the intervention.

Lee, YH. et al. conducted a study in 54 children with obesity to evaluate the effects of aerobic and combined aerobic plus resistance exercise on multiple health parameters, including ALT [28]. There was no minimum ALT value required for study eligibility. The baseline ALT

**Table 2. Exercise protocols and results of included studies.**

| Study | Groups by exercise type | Treatment duration, weeks | Frequency, n/week | Session duration, min | Outcome | Baseline, mean | Post-treatment, mean |
|---|---|---|---|---|---|---|---|
| **RCT** | | | | | | | |
| Lee, YH. 2010 [28] | Combined[a] | 10 | 3 | 60 | ALT | 17 U/L | 18 U/L |
| | Aerobic | 10 | 3 | 60 | ALT | 53 U/L | 67 U/L |
| | Control[b] | N/A | N/A | N/A | ALT | 30 U/L | 38 U/L |
| Davis 2011 [30] | Combined[a] | 16 | 2 | 60–90 | HFF | 6.4% | — |
| | Control[b] | N/A | N/A | N/A | HFF | 8.2% | — |
| Hasson 2012 [31] | Resistance[c] | 16 | 2 | 60 | HFF | — | — |
| | Control[c,d] | N/A | N/A | N/A | HFF | — | — |
| De Piano 2012 [32] | Aerobic | 52 | 3 | 60 | ALT | 46 U/L | 39 U/L |
| | | | | | GGT | 27 U/L | 22 U/L |
| | Combined[a] | 52 | 3 | 60 | ALT | 24 U/L | 20 U/L |
| | | | | | GGT | 19 U/L | 17 U/L |
| Lee, S. 2012 [33] | Aerobic | 12 | 3 | 60 | MRS | 4.7% | 3.7% |
| | Resistance | 12 | 3 | 60 | MRS | 2.9% | 1.8% |
| | Control[e] | N/A | N/A | N/A | MRS | 2.2% | 3.1% |
| Lee, S. 2013 [34] | Aerobic | 12 | 3 | 60 | MRS | 2.2% | 1.3% |
| | Resistance | 12 | 3 | 60 | MRS | 2.0% | 2.1% |
| | Control[f] | N/A | N/A | N/A | MRS | 3.0% | 3.8% |
| Hay 2016 [35] | High-intensity | 24 | 3 | 30-45[g] | MRS | 6.4% | 6.1% |
| | Moderate-intensity | 24 | 3 | 30-45[g] | MRS | 5.8% | 7.3% |
| | Control[e] | N/A | N/A | N/A | MRS | 5.7% | 7.4% |
| De Lira 2017 [36] | Low-intensity | 12 | 3 | Differed[g] | ALT | 20 U/L | 20 U/L |
| | High-intensity | 12 | 3 | Differed[g] | ALT | 26 U/L | 23 U/L |
| Lee, S. 2019 [37] | Aerobic | 24 | 3 | 60 | MRS | 2.5% | 1.9% |
| | Resistance | 24 | 3 | 60 | MRS | 3.0% | 2.7% |
| | Combined[a] | 24 | 3 | 60 | MRS | 2.2% | 1.6% |
| González-Ruíz 2022 [40] | High-intensity | 24 | 4 | 60 | CAP | 222 dB/m | 202 dB/m |
| | | | | | ALT | 17 U/L | 16 U/L |
| | | | | | GGT | 18 U/L | 19 U/L |
| | Low-intensity | 24 | 4 | 60 | CAP | 240 dB/m | 234 dB/m |
| | | | | | ALT | 22 U/L | 23 U/L |
| | | | | | GGT | 22 U/L | 24 U/L |
| | High- and low-intensity | 24 | 4 | 60 | CAP | 218 dB/m | 202 dB/m |
| | | | | | ALT | 15 U/L | 14 U/L |
| | | | | | GGT | 20 U/L | 18 U/L |
| | School-based | 24 | 1 | 60 | CAP | 220 dB/m | 224 dB/m |
| | | | | | ALT | 19 U/L | 21 U/L |
| | | | | | GGT | 16 U/L | 16 U/L |
| Tas 2023 [25] | High-intensity | 4 | 3 | 45 | MRI-PDFF | 5.3% | 5.0% |
| | | | | | ALT | 20 U/L | 20 U/L |
| | Control[e] | N/A | N/A | N/A | MRI-PDFF | 9.0% | 8.2% |
| | | | | | ALT | 30 U/L | 29 U/L |
| **NRCT** | | | | | | | |

(*Continued*)

**Table 2.** (Continued)

| Study | Groups by exercise type | Treatment duration, weeks | Frequency, *n*/week | Session duration, min | Outcome | Baseline, mean | Post-treatment, mean |
|---|---|---|---|---|---|---|---|
| Labayen 2020 [38] | Combined[a] | 22 | 3 | 90 | HFF | 5.6% | 4.5% |
| | | | | | GGT | 17 U/L | 15 U/L |
| | Control[h] | N/A | N/A | N/A | HFF | 5.2% | 5.2% |
| | | | | | GGT | 15 U/L | 15 U/L |
| Iraji 2021 [39] | School-based | 8 | 3 | 50–60 | ALT | 40 U/L | 35 U/L |
| | High-intensity | 8 | 3 | 36–40 | ALT | 37 U/L | 33 U/L |
| | Control[i] | N/A | N/A | N/A | ALT | 37 U/L | 38 U/L |
| **Uncontrolled trial** | | | | | | | |
| Bell 2007 [26] | Combined[a] | 8 | 3 | 60 | ALT | 26 U/L | 26 U/L |
| Van der Heijden 2010 (a) [27] | Aerobic | 12 | 4 | 50 | MRS[j] | 9% | 6% |
| | | | | | ALT | 39 U/L | 35 U/L |
| Van der Heijden 2010 (b) [29] | Resistance | 12 | 2 | 60 | MRS | 9.2% | 9.4% |

See S4 File for detailed descriptions of interventions. Abbreviations: RCT, randomized controlled trial; MRS, magnetic resonance spectroscopy; N/A, not applicable; CAP, controlled attenuation parameter; ALT, alanine transaminase; GGT, gamma-glutamyltransferase; HFF, hepatic fat fraction;—, no data available; NRCT, non-randomized controlled trial.

[a]Combined aerobic and resistance training.

[b]The control group received no intervention.

[c]No baseline values were reported for the intervention groups. The baseline mean hepatic fat fraction for the entire study population was 6.1%.

[d]The control group received carbohydrate nutrition education once per week and a total of four motivational interviewing sessions. The resistance exercise group received the same intervention in addition to strength training twice per week.

[e]The control group was instructed to maintain their current leisure activities and refrain from beginning an exercise program.

[f]The control group was asked not to participate in structured physical activities. All participants were also asked to follow a weight-maintenance diet.

[g]Differed based on energy expenditure of 350 kcal

[h]The control group attended 45-minute lifestyle sessions and 45-minute psycho-educational sessions once every two weeks for 22 weeks. The exercise group had the same 22-week program in addition to the exercise intervention.

[i]The control group did not have a specified intervention.

[j]Values approximated from bar graph.

values were normal for the control group and the combined exercise group, while they were elevated for the aerobic group. Both the aerobic and the combined exercise groups partook in a structured regimen of three weekly 60-minute sessions over a 10-week period. Mean ALT increased in the control group and did not change in the aerobic group or the combined exercise group.

Hay et al. conducted a RCT in 106 adolescents aged 13 to 19 years with BMI ≥ 85th percentile to evaluate the effects of physical activity intensity on insulin sensitivity [35]. Liver fat was a secondary outcome and was measured by MRS. Adolescents were not required to have hepatic steatosis in order to participate. Consequently, only 35 of the 106 (33%) adolescents had hepatic steatosis at baseline. Participants were randomized to one of three treatment groups: high-intensity exercise, moderate-intensity exercise, or control. Because the majority of participants did not have NAFLD, the baseline liver fat values for each group were only slightly higher than normal. Both exercise groups were prescribed 30 to 45 minutes of exercise three times each week for a total of 24 weeks. However, there was only ~60% attendance at the prescribed exercise sessions between the intervention arms. There were no changes seen in mean liver fat for any of the three groups.

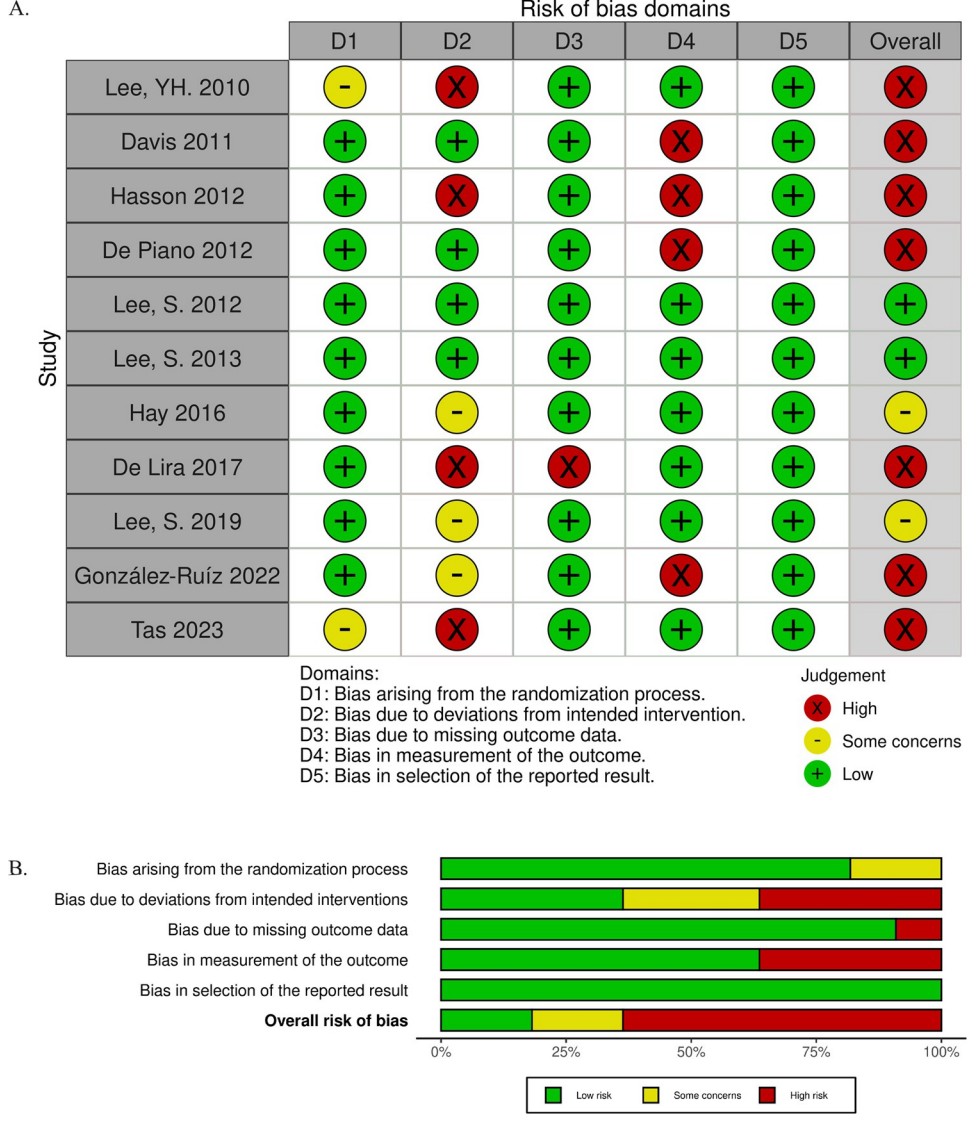

**Fig 2. RoB 2 assessments.** (A) Domain-level judgments. (B) Distribution of risk-of-bias judgments within each bias domain.

Tas et al. conducted a RCT involving 40 adolescents with obesity to evaluate the effects of short-term HIIT on liver fat and cardiometabolic markers [25]. To improve the efficiency of the clinical trial, they used the controlled attenuation parameter (CAP) score as the inclusion criterion intended to assure the presence of steatosis. However, CAP resulted in a low positive predictive value of 42.5%, leading to the inclusion of many participants without MASLD. Consequently, the majority of adolescents did not actually have the condition. Moreover, the baseline ALT level was within the normal range, with a mean of 22 U/L. The study found no significant overall reduction in intrahepatic triglyceride content for the HIIT group ($\Delta$ = -0.31 percentage points, 95% CI: -0.77 to 0.15; p = 0.179). However, a post hoc analysis restricted to those with MASLD revealed a modest decrease in MRI PDFF of -1.05 percentage points ($\Delta$ = -1.05 percentage points, 95% CI: -2.08 to -0.01; p = 0.048).

A.

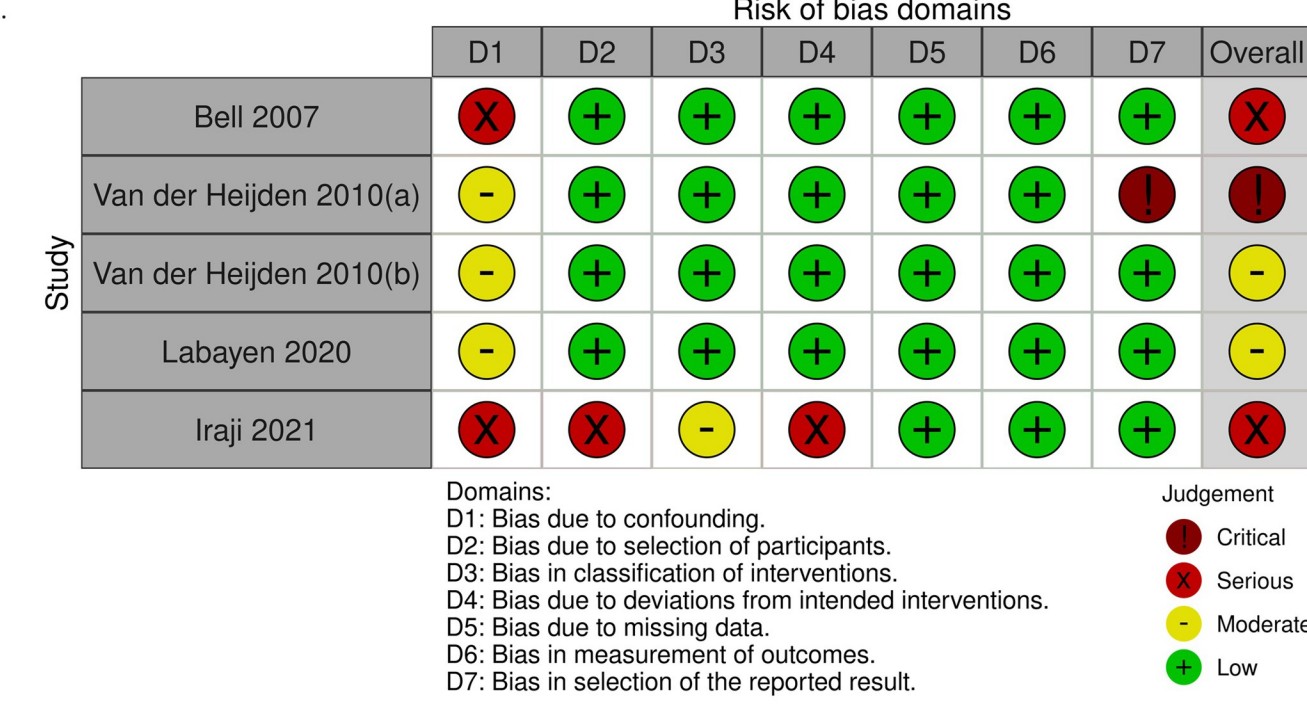

B.

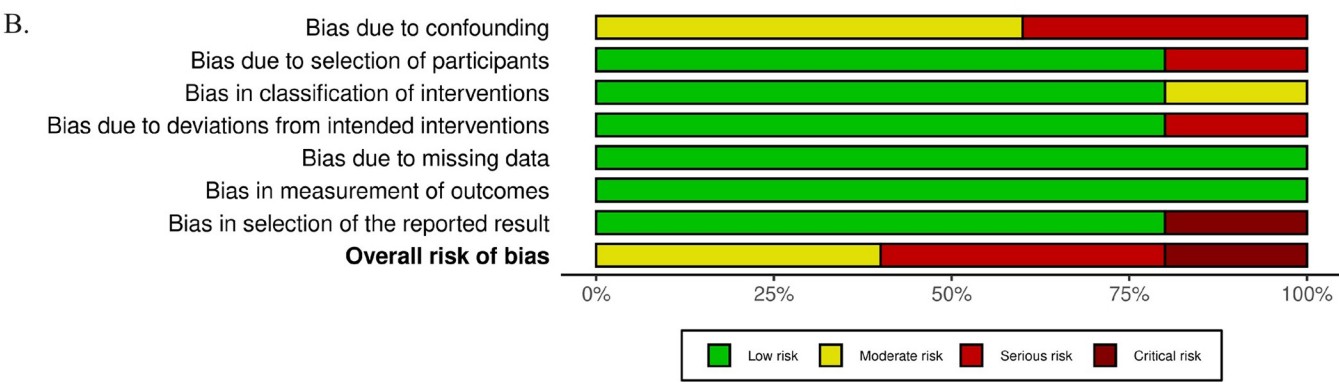

**Fig 3. ROBINS-I assessments.** (A) Domain-level judgments. (B) Distribution of risk-of-bias judgments within each bias domain.

**Uncontrolled studies of exercise interventions.** Van der Heijden et al. conducted an uncontrolled study to explore the influence of aerobic exercise on hepatic fat content and ALT levels within a group of 15 Hispanic adolescents with obesity. Importantly, it is worth noting that only five participants displayed evidence of hepatic steatosis at the outset of the study [27]. The prescribed exercise program consisted of four sessions per week, each lasting 50 minutes, on a treadmill, elliptical, or bicycle, over a 12-week period. Mean hepatic steatosis, measured by MRS, decreased from approximately 9% to approximately 6%. Mean ALT did not change significantly.

In another uncontrolled study by van der Heijden et al., 12 Hispanic adolescent males with obesity received strength training consisting of two sessions of 60 minutes each per week [29].

Liver fat was measured by MRS before and after 12 weeks of intervention. Hepatic steatosis was not required but was present in 7 of the 12 participants. Liver fat did not change with the intervention in adolescents with high liver fat at baseline (13.9% vs. 14.2%) or those with normal liver fat at baseline (2.7% vs. 2.7%).

In an uncontrolled trial by Bell et al., 14 sedentary children with obesity participated in an 8-week circuit training program involving aerobic and resistance exercises [26]. The program consisted of three 60-minute sessions per week. The investigators assessed ALT before and after the intervention. Children were not required to have elevated ALT in order to participate. At baseline, the mean ALT was close to normal (26 U/L) and did not change with the intervention.

## Exercise interventions added to other lifestyle interventions

We identified two controlled trials examining the effects of exercise in addition to lifestyle interventions on hepatic fat and chemistries. Labayen et al. conducted a study focused on children with overweight, investigating the effects of a family-based lifestyle intervention, either with or without an exercise component, on hepatic fat, adiposity, and cardiometabolic risk factors [38]. The study did not require the presence of hepatic steatosis or elevated liver chemistry for participation. Among the participants, 36% had hepatic steatosis. In the control group, children and their caregivers separately attended 45-minute lifestyle sessions and 45-minute psycho-educational sessions once every two weeks, accumulating to a total of 11 sessions over 22 weeks. The exercise group had the same 22-week program in addition to three weekly 90-minute sessions of game-based cardiovascular and strength exercises. Because most children did not have hepatic steatosis, the mean liver fat at baseline in the exercise group was close to normal, 5.6%, and decreased to 4.5% following the intervention. The mean liver fat at baseline in the control group was 5.2% and did not change with the intervention. Additionally, mean GGT levels were normal at baseline and remained normal following the intervention in the exercise (17.0 and 15.4) and control (15.3 and 15.4) groups.

A study conducted by Hasson et al. explored whether African American and Hispanic adolescents with obesity would respond differently to a 16-week intervention designed to reduce adiposity and low-grade inflammation and improve insulin sensitivity [31]. Within a cohort of 126 participants, the study examined the change in liver fat as a secondary outcome by assessing MRI hepatic fat fraction among those who successfully completed the study (n = 100). Importantly, the study's inclusion criteria did not necessitate the presence of hepatic steatosis, leading to the enrollment of participants predominantly with a normal hepatic fat fraction at baseline. The nutrition group received a 16-week dietary intervention focused on reducing calorie intake from added sugar and increasing fiber intake, along with four individual motivational interviewing sessions. The exercise plus nutrition group completed two strength training sessions (60 minutes per session) per week for 16 weeks, in addition to the same intervention as the nutrition group. There was no change in hepatic fat fraction in the nutrition group. In the exercise plus nutrition group, there was an absolute decrease in the mean hepatic fat fraction of 1.3%.

## Studies comparing an exercise intervention with another exercise intervention

Our literature search identified nine controlled trials that compared the effects of different types of exercise on liver fat and chemistries in children with or at risk for NAFLD due to overweight or obesity. Lee, S. et al. conducted a 24-week RCT with 118 adolescents with overweight or obesity to compare the effects of aerobic exercise, resistance exercise, and a combined regimen of

aerobic and resistance exercise on insulin sensitivity [37]. Liver fat was evaluated as a secondary outcome. Notably, participants were not required to have hepatic steatosis, and the average liver fat content was found to be within the normal range across all groups. Each group exercised three times per week for 60 minutes per session. Liver chemistry started as normal for each group and remained low following the intervention. Changes in liver fat for all groups were minimal—less than a 1% change, well below the precision of MRI-measured liver fat.

González-Ruíz et al. conducted a RCT investigating different exercise interventions in 120 adolescents with obesity [40]. The study objective was to evaluate the effects of distinct school-based physical education exercise programs on estimated liver fat content. Participants were not required to have hepatic steatosis in order to participate. Liver fat was estimated by CAP measurement with VCTE. Notably, the mean CAP value at baseline was normal, signaling that the majority of the study participants did not have hepatic steatosis. Baseline levels of both ALT and GGT were within the normal range for the majority of participants. The study included four groups: standard school-based physical education, low-intensity physical education, high-intensity physical education, and a combination of low- and high-intensity physical education. The standard group engaged in 60 minutes of low-to-moderate intensity physical activity once a week for 24 weeks. In contrast, the low-intensity, high-intensity, and combination groups participated in three weekly 60-minute exercise sessions, augmented by an extra standard session, each sustained over a 24-week period. Mean CAP values did not change in the standard or the low-intensity groups. Mean CAP values decreased by 20.02 dB/m in the high-intensity group and 16.25 dB/m in the combination group. Moreover, the levels of ALT and GGT exhibited no significant changes across any of the groups.

De Piano et al. studied 58 children with obesity, of which 28 had abnormal liver ultrasound [32]. All 58 children were randomized to aerobic training or aerobic plus resistance training according to NAFLD diagnosis. Each group had 3 sessions lasting 60 minutes per week over the course of 52 weeks. Both groups also received psychological and nutrition education once a week. The study objective was to evaluate the change in liver chemistry, although there was no minimum value required for ALT or GGT in order to participate. Serum ALT was elevated at baseline in the aerobic group (mean 46 U/L) and normal in the combined exercise group (mean 24 U/L). Serum GGT was normal in both groups. There were no significant changes in ALT or GGT levels for either group.

De Lira et al. conducted an exercise study in 51 adolescents with obesity and compared the effects of high-intensity training versus low-intensity training on ALT levels [36]. Adolescents were not required to have elevated ALT to participate. Consequently, most participants had normal liver chemistry at baseline. The 12-week exercise regimen consisted of three treadmill sessions per week, with each tailored to burn 350 kcal. The high-intensity group exercised at ventilatory threshold 1 (VT1), and the low-intensity group exercised at an intensity 20% below VT1. Additionally, both groups received 1-hour weekly nutrition and psychological counseling sessions. At baseline, the mean ALT levels were within the normal range for both the high-intensity group (26 U/L) and the low-intensity group (20 U/L), and these values exhibited no significant changes following the intervention.

In the 'Exercise only interventions' section of the results, five studies comparing different exercise modalities were previously described [28, 33–35, 39].

## Selected studies to assess the clinical context of the results presented in this review

As described in the methods, we examined the study populations in clinical trials of children with biopsy-proven NAFLD to understand the baseline liver-related measures in these

children. Table 3 provides clinical and demographic data for seven study cohorts comprising a total of 599 children with NAFLD. These children had baseline mean liver fat percentages, measured by MRI PDFF, ranging from 14% to 23%. Baseline mean ALT and GGT ranged from 46 U/L to 123 U/L and 21 U/L to 52 U/L, respectively [5, 41–47].

In contrast, the studies included in our review had baseline mean hepatic fat fractions ranging from 2% to 9%. Additionally, the baseline mean ALT and GGT values ranged from 18 U/L to 39 U/L and 16 U/L to 21 U/L, respectively. These findings are illustrated in Table 1.

## Discussion

We performed a systematic review of the extant literature to evaluate the effects of exercise intervention on hepatic steatosis and liver chemistry in children with NAFLD and/or MASLD. Our systematic search identified 16 studies that specifically evaluated the impact of exercise on liver health. Among the 11 studies that evaluated hepatic steatosis, five reported a modest absolute decrease, ranging from 1% to 3%, following the intervention. Notably, only seven of these 11 studies enrolled populations with a mean hepatic steatosis $\geq$ 5% at baseline, and among these, three showed an absolute decrease in hepatic steatosis of 1% to 3% after the intervention. The types of exercise interventions in the studies that demonstrated a decrease in hepatic steatosis varied and included aerobic, resistance, and combined aerobic and resistance training. Similarly, only six studies had an elevated mean baseline ALT value, and these were all mild, with levels ranging from 26 to 39 units per liter (U/L). Furthermore, none of the studies featured elevated mean baseline GGT values. Because these laboratory values were so close to

**Table 3. Participant characteristics in studies of children with biopsy-proven NAFLD.**

| Study | Study sample size | Age, years | Female sex, % | ALT, U/L | AST, U/L | GGT, U/L | Liver fat measure, % | Steatosis grade, % or mean (SD) | Participants with NASH, n (%) |
|---|---|---|---|---|---|---|---|---|---|
| Lavine 2011 [41] | 173 | 13.1 | 19 | 123 | 71 | 51 | — | 2.2 (0.8) | 73 (42%) |
| Pacifico 2015 [42] | 51 | 10.9 | 41 | 56 | — | — | 14.0–15.5[a] | Grade 1: 28 Grade 2: 33 Grade 3: 39 | 33 (65%) |
| Della Corte 2016 [43] | 43 | 12.8 | 56 | 46 | 31 | 21 | — | Grade 1: 21 Grade 2: 49 Grade 3: 30 | 14 (33%) |
| Zöhrer 2016 [44] | 40 | 13.2 | 40 | 52 | 35 | — | — | Grade 1: 37.5 Grade 2: 40 Grade 3: 22.5 | 40 (100%) |
| Schwimmer 2017 [45] | 169 | 13.7 | 30 | 123 | 71 | 47 | 21.1 (9.8)[b,c] | 2.4 (0.7) | 44 (26%) |
| Schwimmer 2019 [5] | 40 | 13.1 | 0 | 73–82[d] | 39–44[e] | 42–53[f] | 23 (9.7)[c] | — | 11 (28%) |
| Vos 2022 [46] | 83 | 13 | 19 | 120 | 64 | 52 | — | Grade 1: 12 Grade 2: 29 Grade 3: 59, 2.5 (0.7) | 20 (24%) |

Abbreviations: ALT, alanine transaminase; AST, aspartate aminotransferase; GGT, gamma-glutamyltransferase; NASH, non-alcoholic steatohepatitis;—, no data available.

[a]Hepatic fat fraction assessed by MRI. The median hepatic fat fraction in the intervention and placebo groups ranged from 14.0–15.5%.

[b]MRI-PDFF available for 110 participants.

[c]Hepatic steatosis assessed by MRI-PDFF.

[d]The median ALT in the intervention and placebo groups ranged from 73–82 U/L.

[e]The median AST in the intervention and placebo groups ranged from 39–44 U/L.

[f]The median GGT in the intervention and placebo groups ranged from 42–53 U/L.

normal to begin with, there was minimal opportunity for improvement in ALT or GGT. Consequently, no substantial changes in liver chemistry were observed across the spectrum of studies incorporated in this review. While this systematic review lays the foundation for evaluating the evidence-based utility of exercise interventions for treating MASLD in pediatric populations, it also underscores the necessity for further investigation.

## Gaps in study inclusion

The results of our systematic review shed light on a prominent limitation within the existing body of research regarding exercise's impact on hepatic steatosis and liver chemistry in children with NAFLD and/or MASLD. Notably, the majority of available studies were not explicitly designed for the assessment of children with NAFLD. None of these studies originated from clinical populations comprising patients with NAFLD. Moreover, many of the studies did not explicitly ensure that all participants had steatotic liver disease. Instead, many of the studies were primarily geared towards investigating changes in insulin resistance in children with obesity, with liver fat content and liver chemistry only included as secondary outcomes. Since no studies included patients with liver histology, the overall severity of disease in the study populations was unknown (Table 4). Furthermore, in the studies that measured liver fat through imaging or liver chemistry, the mean values tended to be normal or mildly elevated. Baseline mean liver fat ranged from 2% to 9%, and ALT levels ranged from 18 U/L to 39 U/L (Table 1). This contrasts sharply with the typical presentation of patients with NAFLD, who typically exhibit a high degree of steatosis with inflammation and fibrosis [4]. Our examination of clinical trials for pediatric NAFLD that enrolled children with biopsy-proven NAFLD revealed baseline mean magnetic resonance imaging-proton density fat fraction (MRI-PDFF) ranging from 14% to 23% and ALT levels ranging from 46 U/L to 123 U/L (Table 3). Although ALT levels in clinical populations, especially those who undergo liver biopsy, skew higher, the fact that the ranges of the reported exercise studies and clinical trials do not overlap further demonstrates that they represent substantially different populations of children. To accurately assess the impact of exercise as a treatment, clinical trials must include children who reflect the typical patient with NAFLD and/or MASLD and, thus, the individuals most in need of treatment. Consequently, the study designs and participant recruitment approaches used in the selected studies limit the ability to apply the study findings to individual patients with NAFLD and/or MASLD.

## Use of validated assessment tools

To effectively gauge the response to an intervention in pediatric MASLD, clinical trials must use an outcome measure that is quantitative, sensitive to change, and validated longitudinally in children with MASLD. While liver biopsy is the clinical reference standard for diagnosing hepatic steatosis, its invasive nature limits its use in longitudinal clinical trials [12, 13, 48]. Among the non-invasive imaging alternatives, both MRI and MRS are acknowledged for their reproducibility, safety, and accuracy. The emergence of MRI-PDFF has further elevated the precision of liver fat quantification, mitigating confounders present in earlier MRI methods. Both MRS and MRI-PDFF have proven accuracy in measuring liver steatosis in children and have validated diagnostic cut-off values [12–16]. Conversely, CAP lacks validation as a measure of change and lacks defined cut-off values in children [12, 17]. Additionally, while liver chemistry alone is not reliable for quantifying steatosis, variations in ALT and GGT levels have shown a relationship to histological improvements in children with steatotic liver disease [15, 21]. Importantly, serum ALT remains the most sensitive and specific biochemical marker of hepatocellular injury and inflammation related to steatotic liver disease [21].

**Table 4. Study populations of included studies.**

| Study | Study sample size | Did the study population have NAFLD? | Did the study population have NASH? |
|---|---|---|---|
| Bell 2007 [26] | 14 | No | Unknown |
| Van der Heijden 2010(a) [27] | 15 | Less than half (33%) | Unknown |
| Lee, YH. 2010 [28] | 54 | Not stated[a] | Unknown |
| Van der Heijden 2010(b) [29] | 12 | More than half (58%) | Unknown |
| Davis 2011 [30] | 38 | Not stated[b] | Unknown |
| Hasson 2012 [31] | 100 | Not stated[c] | Unknown |
| De Piano 2012 [32] | 58 | Possibly[d] | Unknown |
| Lee, S. 2012 [33] | 45 | No | Unknown |
| Lee, S. 2013 [34] | 44 | No | Unknown |
| Hay 2016 [35] | 106 | Less than half (33%) | Unknown |
| De Lira 2017 [36] | 84 | No | Unknown |
| Lee, S. 2019 [37] | 118 | No | Unknown |
| Labayen 2020 [38] | 116 | Less than half (36%) | Unknown |
| Iraji 2021 [39] | 34 | Possibly[e] | Unknown |
| González-Ruíz 2022 [40] | 120 | No | Unknown |
| Tas 2023 [25] | 37 | Less than half (41%) | Unknown |

See Table 3 for reference values. Abbreviations: NAFLD, nonalcoholic fatty liver disease; NASH, non-alcoholic steatohepatitis.

[a]There was no minimum ALT value required for study eligibility. The study population had a mean ALT of 32 U/L.

[b]Participants were not required to have hepatic steatosis to participate. The mean baseline hepatic fat fraction was slightly elevated (6.81%).

[c]Participants were not required to have hepatic steatosis to participate. The mean baseline hepatic fat fraction was slightly elevated (6.08%).

[d]Participants were required to have an abnormal liver ultrasound. The mean baseline ALT was 30.22 U/L.

[e]Participants were required to have hepatic ultrasound grade 1 or 2 and ALT > 25.8 U/L.

While MRS and MRI-PDFF currently stand as the most accurate and validated non-invasive techniques for quantifying hepatic steatosis, it is noteworthy that only six studies within this review utilized MRS to assess hepatic steatosis, and one employed MRI-PDFF. Additionally, only four studies used both liver imaging and chemistry to more comprehensively assess the presence and severity of NAFLD. Instead, many of the studies in this review relied on MRI hepatic fat fraction or solely on liver chemistry. The lack of validated assessment tools used in these studies raises concerns about the reliability of their results. This suggests caution in assuming that these findings fully represent the impact of exercise on MASLD in children.

## Meaningful changes in MASLD parameters

When designing an intervention to decrease hepatic steatosis, the observed change must be greater than the natural fluctuations in steatosis. The STEATOSIS study, for instance, established that an absolute change of MRI-PDFF $\geq$ 4% is necessary for clinical significance in adolescents with NASH over a 12-week timeframe [14]. Similarly, in adults with NASH over a 72-week period, a 30% absolute reduction in MRI-PDFF is associated with histological improvement in NASH [18]. To meet these benchmarks, clinical trials must enroll children with high enough liver fat to be able to demonstrate meaningful improvement. In addition to steatosis, changes in serum ALT and GGT are associated with improvements in inflammation and fibrosis in children with steatotic liver disease [21]. Similar to the issue with steatosis, clinical trials must enroll children with high enough ALT and GGT levels to both demonstrate the presence of active disease and allow for the potential to demonstrate clinically meaningful change. For example, the studies referenced in Table 3 had a mean baseline MRI-PDFF

ranging from 14% to 23%, ALT between 46 U/L and 123 U/L, and GGT between 21 U/L and 53 U/L. It is expected that future trials of exercise as a treatment for MASLD would have similar baseline values.

### Existing literature and current recommendations

Understanding the effects of exercise on pediatric MASLD is crucial because lifestyle modifications, including increased physical activity, represent the initial therapeutic approach for all children diagnosed with MASLD [2]. Physical activity plays an important role in children's health and development. The American Academy of Pediatrics currently recommends at least 60 minutes of daily moderate to vigorous physical activity for children aged 6–17 years [49]. Nevertheless, its effectiveness as a targeted treatment for MASLD remains uncertain. While the evidence base in pediatrics is limited, the American College of Sports Medicine recently issued physical activity recommendations for adults with NAFLD [50]. Over 25 studies in adults with NAFLD have shown that exercise can improve liver fat, and four clinical trials have shown that exercise can improve liver histology [9]. However, the optimal exercise prescription remains unclear due to significant variations in the frequency, intensity, time, and type of exercise studied [9, 51–54]. Given the evolving body of research, exercise holds promise as a viable treatment option for MASLD. The translation of adult-focused evidence to children remains an intriguing yet uncertain prospect. Further investigations are needed to ascertain the applicability of these findings in adults to pediatric populations and to delineate the precise parameters for exercise interventions tailored to children with MASLD.

### Strengths and limitations

The strengths of this systematic review are rooted in our comprehensive and meticulous approach. We conducted a systematic search across six databases and searched the reference lists of primary studies and pertinent reviews, ensuring a thorough examination of the existing literature. Additionally, two review authors independently appraised each study, enhancing the quality of our analysis. Another strength is our specific focus on the effects of exercise intervention alone on quantifiable clinical parameters related to MASLD.

Several limitations of this review were also identified. First, the exclusion of non-English and unpublished studies may introduce language and publication bias, potentially narrowing the diversity of data and limiting generalizability. Additionally, due to the limited number of studies on this topic, we included non-randomized and uncontrolled studies to provide a broader perspective, which may introduce confounding variables and increase the risk of bias. Furthermore, most of the included studies exhibited a high risk of bias, impacting the reliability of the derived conclusions. As detailed in the discussion, it is essential to interpret the results of these studies cautiously, underscoring the need for additional research on this topic. Additionally, our inability to access individual-level data for these studies restricted our capacity to conduct more detailed analyses. Finally, the collective body of evidence did not support a meta-analysis, limiting our ability to provide a quantitative summary of the available data.

### Future directions

This review highlights the need for rigorous future studies to examine the effects of exercise on pediatric MASLD and to establish the most effective exercise guidelines for affected children. Fig 4 outlines research priorities intended to advance the field toward the development of clinically valuable, evidence-based exercise recommendations for children with MASLD [9].

**Study Design and Outcome Measures:**

1. Conduct studies in children with clinically documented MASLD.
2. Conduct clinical trials using outcome measures that are quantitative, sensitive, and longitudinally validated in children with MASLD. Currently, these are ALT, GGT, MRI-PDFF, and liver histology.
3. Perform studies that are adequately powered to demonstrate clinically meaningful changes in liver-related outcomes.

**Optimization of Exercise Interventions:**

4. Determine the dose of exercise needed to achieve clinically relevant benefits in children with MASLD.
5. Identify the most beneficial exercise regimens for children with MASLD, taking into account the type, intensity, frequency, and duration of exercise.
6. Investigate the development of personalized exercise prescriptions for children with MASLD.

**Role of Exercise in MASLD Pathogenesis:**

7. Explore the physiological mechanisms underlying exercise's suggested benefits in children with MASLD.

**Long-Term Outcomes and Adherence to Exercise:**

8. Evaluate whether sustained physical activity improves long-term health outcomes, such reducing cardiovascular disease, liver and extrahepatic cancers, cirrhosis, and mortality in children with MASLD.
9. Investigate strategies to improve adherence and long-term commitment to exercise regimens among children with MASLD.

**Combined Therapeutic Approaches and Pharmacologic Interventions:**

10. Explore the potential additive effect of combining exercise with other modalities such as diet, medication, or supplements.

**Fig 4. Future research directions.**

## Conclusions

There is limited evidence on the effects of exercise intervention in treating children with MASLD. This lack of evidence likely reflects the methodological limitations of the current evidence base. Given that increased physical activity is a main treatment strategy for pediatric MASLD, developing precise, evidence-based exercise guidelines is essential for the effective clinical management of this condition. High-quality future studies are needed for exercise intervention to translate to clinical care for children with MASLD.

## Supporting information

**S1 File. Review protocol.**
(DOCX)

**S2 File. Detailed electronic search strategy.**
(DOCX)

**S3 File. Detailed methods for assessing risk of bias in included studies.**
(DOCX)

**S4 File. Detailed descriptions of interventions.**
(DOCX)

**S5 File. PRISMA 2020 checklist.**
(DOCX)

**S6 File. PRISMA 2020 abstract checklist.**
(DOCX)

**S7 File.**
(CSV)

**S8 File.**
(XLSX)

**S9 File.**
(XLSX)

## Author Contributions

**Conceptualization:** Martha R. Smith, Karen M. Heskett, Jeffrey B. Schwimmer.

**Data curation:** Martha R. Smith.

**Investigation:** Martha R. Smith, Elizabeth L. Yu.

**Methodology:** Martha R. Smith.

**Supervision:** Karen M. Heskett, Jeffrey B. Schwimmer.

**Validation:** Ghattas J. Malki.

**Visualization:** Martha R. Smith, Ghattas J. Malki.

**Writing – original draft:** Martha R. Smith, Ghattas J. Malki, Karen M. Heskett, Jeffrey B. Schwimmer.

**Writing – review & editing:** Martha R. Smith, Elizabeth L. Yu, Ghattas J. Malki, Kimberly P. Newton, Nidhi P. Goyal, Karen M. Heskett, Jeffrey B. Schwimmer.

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
