## [Decision Letter · Decision Letter 0]

23 Jul 2024

PONE-D-24-08084Systematic review of exercise for the treatment of pediatric metabolic dysfunction-associated steatotic liver diseasePLOS ONE

Dear Dr. Schwimmer,

Thank you for submitting your manuscript to PLOS ONE. After careful consideration, we feel that it has merit but does not fully meet PLOS ONE’s publication criteria as it currently stands. Therefore, we invite you to submit a revised version of the manuscript that addresses the points raised during the review process.

We look forward to receiving your revised manuscript.

Kind regards,

Shaghayegh Khanmohammadi

Academic Editor

PLOS ONE

Additional Editor Comments:

1. Ensure the manuscript is free of English language errors. You can use language-editing services.

Reviewers' comments:

Reviewer's Responses to Questions

**Comments to the Author**

1. Is the manuscript technically sound, and do the data support the conclusions?

Reviewer #1: Yes

Reviewer #2: Partly

2. Has the statistical analysis been performed appropriately and rigorously? 

Reviewer #1: N/A

Reviewer #2: Yes

3. Have the authors made all data underlying the findings in their manuscript fully available?

Reviewer #1: Yes

Reviewer #2: Yes

4. Is the manuscript presented in an intelligible fashion and written in standard English?

Reviewer #1: Yes

Reviewer #2: No

5. Review Comments to the Author

Reviewer #1: In this systematic review, Smith and colleagues have collected and summarized controlled and non-controlled exercise studies for the treatment of MASLD/NAFLD. Strengths include a rigorous application of the systematic review process and clear presentation of the results. Their findings point towards the lack of high-quality studies in this field, which is important in its own right.

One interesting results is that most studies in this field did not require patients to have been diagnosed with NAFLD, thereby introducing heterogeneity within the groups, and lowering the chance of finding significant results. The authors compare the patient characteristics to published cohorts of pediatric NAFLD patients. While this reinforces this point, this step is a bit superfluous in my opinion. Especially comparing serum ALT levels is not straightforward, as biopsy-proven pediatric cohorts themselves are subject to sampling bias (at the other end of the spectrum), as reflected in quite high ALT levels in some cohorts. If the authors choose to retain this comparison, this caveat should be clearly mentioned.

Minor points:

• The most common risk of bias was ‘deviation from the intended interventions’. The reader would be well served if the specifics of this could be discussed in a bit more detail.

• Every study is discussed separately, without presenting a broader view in the results section. I would for instance discuss the two papers by Lee S et al. together.

Reviewer #2: I recommend major revision. Meta-regression analysis for confounding variables such as age, gender and past medical history. There are so many factors that can affect the result, especially the history of exercise. These should be mentioned in the paper. Also, the search strategy must be updated. It is for one year ago.

6. PLOS authors have the option to publish the peer review history of their article (what does this mean?). If published, this will include your full peer review and any attached files.

Reviewer #1: No

Reviewer #2: **Yes: **Mobina Fathi

---

## [Author Response · Author response to Decision Letter 0]

19 Aug 2024

Reviewer #1

In this systematic review, Smith and colleagues have collected and summarized controlled and non-controlled exercise studies for the treatment of MASLD/NAFLD. Strengths include a rigorous application of the systematic review process and clear presentation of the results. Their findings point towards the lack of high-quality studies in this field, which is important in its own right.

1. One interesting results is that most studies in this field did not require patients to have been diagnosed with NAFLD, thereby introducing heterogeneity within the groups, and lowering the chance of finding significant results. The authors compare the patient characteristics to published cohorts of pediatric NAFLD patients. While this reinforces this point, this step is a bit superfluous in my opinion. Especially comparing serum ALT levels is not straightforward, as biopsy-proven pediatric cohorts themselves are subject to sampling bias (at the other end of the spectrum), as reflected in quite high ALT levels in some cohorts. If the authors choose to retain this comparison, this caveat should be clearly mentioned.

Response: We appreciate the reviewer’s insightful comments. The utility of evaluating any given intervention for MASLD/NAFLD is deeply tied to the relevance of the patient population studied. Given the substantial resources required for clinical trials—both in terms of cost and participant commitment—it is essential that these studies focus on the appropriate populations. Otherwise, their potential to inform clinical practice is diminished. We believe it is crucial for the field to recognize this issue, particularly since many studies have not adequately focused on the relevant populations. Therefore, we consider the inclusion of this comparison necessary rather than superfluous.

We included information on clinical trials involving other interventions to highlight the discrepancy between the clinical populations most in need of treatment and those typically enrolled in exercise studies. We agree that patients with MASLD diagnosed in specialty clinics often have more severe disease, which underscores their need for effective treatments. We have added further discussion in the section titled “Gaps in study inclusion” (lines 587-592) to clarify our rationale for this comparison and to address the elevated ALT levels observed in biopsy-proven pediatric cohorts.

Minor points:

2. The most common risk of bias was ‘deviation from the intended interventions’. The reader would be well served if the specifics of this could be discussed in a bit more detail.

Response: We agree that a more detailed discussion on the high risk of bias due to ‘deviation from the intended interventions’ would benefit the reader. We have expanded our commentary on this point in the results section, under ‘Summary of risk of bias assessment’ (lines 299-305), to explain how specific studies deviated from their intended interventions.

3. Every study is discussed separately, without presenting a broader view in the results section. I would for instance discuss the two papers by Lee S et al. together.

Response: We have presented a broader view of the results in the first paragraph of the discussion (lines 551-564). In response to the reviewer’s suggestion, we have rearranged the results section so that the two paragraphs discussing the studies by Lee S et al. are now presented sequentially. However, we believe that discussing these studies separately is important for clarity and consistency with the treatment of other studies in the review.

Reviewer #2

1. I recommend major revision. 

Response: We appreciate the reviewer’s recommendation for major revisions. In response to the reviewers and editors, we updated the search and revised the tables, figures, and text accordingly. 

2. Meta-regression analysis for confounding variables such as age, gender and past medical history. 

Response: The studies included in this review were not suitable for meta-analysis due to the limited data available for each outcome, significant heterogeneity among the included studies in terms of design, population, intervention, and outcome measures, and the high risk of bias present in 10 out of 16 studies. As a result, a meta-analysis was not conducted. We have added a more detailed explanation in the methods section, ‘Data extraction and management’ (lines 198-201), to clarify this decision. We have also cited a relevant methods paper to support our approach. We hope this systematic review will guide future studies on exercise as an intervention for pediatric MASLD, helping them meet the standards necessary for meta-analysis.

3. There are so many factors that can affect the result, especially the history of exercise. These should be mentioned in the paper. 

Response: We revisited all the included studies to address the concern regarding exercise history. Unfortunately, we found that this variable was typically not reported and was not evaluated as a covariate in the outcomes of any of the studies.

4. Also, the search strategy must be updated. It is for one year ago.

Response: In response to your request, we re-ran the original search strategy on August 8, 2024, to ensure the review’s currency. We also included the more recent terms “MASLD” and “metabolic dysfunction-associated steatotic liver disease.” This search identified one additional study that met the inclusion criteria, which has been incorporated into the systematic review. We have updated the PRISMA flow diagram and the relevant sections of the manuscript accordingly.

---

## [Decision Letter · Decision Letter 1]

22 Oct 2024

PONE-D-24-08084R1Systematic review of exercise for the treatment of pediatric metabolic dysfunction-associated steatotic liver diseasePLOS ONE

Dear Dr. Schwimmer,

Thank you for submitting your manuscript to PLOS ONE. After careful consideration, we feel that it has merit but does not fully meet PLOS ONE’s publication criteria as it currently stands. Therefore, we invite you to submit a revised version of the manuscript that addresses the points raised during the review process.

We look forward to receiving your revised manuscript.

Kind regards,

Shaghayegh Khanmohammadi

Academic Editor

PLOS ONE

Reviewers' comments:

Reviewer's Responses to Questions

**Comments to the Author**

1. If the authors have adequately addressed your comments raised in a previous round of review and you feel that this manuscript is now acceptable for publication, you may indicate that here to bypass the “Comments to the Author” section, enter your conflict of interest statement in the “Confidential to Editor” section, and submit your "Accept" recommendation.

Reviewer #1: All comments have been addressed

Reviewer #3: (No Response)

2. Is the manuscript technically sound, and do the data support the conclusions?

Reviewer #1: Yes

Reviewer #3: Yes

3. Has the statistical analysis been performed appropriately and rigorously? 

Reviewer #1: Yes

Reviewer #3: Yes

4. Have the authors made all data underlying the findings in their manuscript fully available?

Reviewer #1: Yes

Reviewer #3: Yes

5. Is the manuscript presented in an intelligible fashion and written in standard English?

Reviewer #1: Yes

Reviewer #3: Yes

6. Review Comments to the Author

**Reviewer #1:** Thank you for updating the manuscript and complying with my (and the other reviewer's) comments. I have no further comments at this point.

**Reviewer #3:** The study in concern possesses considerable merit and focuses on a subject of great significance. However, there are certain points that require clarification before publication.

1. Provide more detailed data on the mentioned prevalence, including demographics such as gender, ethnicity, or nationality, as well as any regional differences to be noted.

2. Has the renaming of NAFLD to MASLD resulted in any modifications to diagnostic criteria or clinical guidelines? If affirmative, what was the impact on your research?

3. The topics addressed in the final paragraph of the introduction pertain to methodology and discussion sections.

4. The inclusion of children with a BMI at or above the 85th percentile is rational but necessitates further examination of its implications. What was the author's rationale for including children with a BMI ≥ 85th percentile, and how does this impact the study's findings? What are the differences in outcomes between overweight and obese children?

5. Elucidate the rationale for incorporating non-randomized and uncontrolled studies, considering their intrinsic bias risk, as this affects the overall quality and conclusions of the review.

6. What was the author's rationale for selecting participants aged 19 years or younger? What criteria did you use to define this age range as children and adolescents?

7. Provide more details on how “higher risk for NAFLD” is defined. Were there any additional risk factors considered beyond overweight and obesity?

8. Explain how the diversity in intervention types (e.g., aerobic vs. resistance training) was managed in the synthesis. Were any subgroup analyses planned or performed?

9. Provide a rationale for selecting the three outcome categories: hepatic steatosis, liver chemistries, and liver histology. Were there any additional outcomes considered and excluded? If so, why?

10. Discuss any limitations of the search strategy, such as the exclusion of non-English studies or grey literature. Were any efforts made to identify unpublished studies?

11. Elucidate the methods employed to rectify discrepancies. Were any particular criteria or thresholds employed?

12. Discuss the implications of finding a high risk of bias in 10 out of 16 studies. How does this impact the overall conclusions and recommendations of the review?

13. Try to avoid bullet points as much as possible.

14. The discussion indicates that merely seven of the 16 studies included populations with a mean hepatic steatosis of ≥ 5%, and only six exhibited elevated baseline ALT levels. This indicates a limited scope that restricts the generalizability of results. Articulate the justification for the inclusion of studies featuring less severe hepatic conditions. How does this inclusion affect the interpretation of exercise's impact on more severe cases of NAFLD/MASLD?

15. The discussion recognizes that negligible alterations in ALT and GGT were noted, attributed to baseline values being nearly normal. Could there be confounding factors or issues with the sensitivity of measurements?

16. Cross-check the data mentioned in the discussion with the results section to ensure consistency.

17. The discussion indicates that numerous studies depend on MRI hepatic fat fraction and liver chemistry. This prompts apprehensions regarding the uniformity and comparability of results. Discuss the implications of using different assessment tools on the comparability of study results. How might these differences affect the overall conclusions drawn from the systematic review?

18. The segment regarding future directions is significant yet deficient in specificity. Offer more comprehensive and precise suggestions for subsequent research endeavors. Which specific study designs, populations, or methodologies should be emphasized?

7. PLOS authors have the option to publish the peer review history of their article (what does this mean?). If published, this will include your full peer review and any attached files.

Reviewer #1: No

Reviewer #3: No

---

## [Author Response · Author response to Decision Letter 1]

5 Nov 2024

Reviewer #1

Thank you for updating the manuscript and complying with my (and the other reviewer's) comments. I have no further comments at this point.

Reviewer #3

The study in concern possesses considerable merit and focuses on a subject of great significance. However, there are certain points that require clarification before publication.

1. Provide more detailed data on the mentioned prevalence, including demographics such as gender, ethnicity, or nationality, as well as any regional differences to be noted.

Response: We appreciate the reviewer’s suggestion to include more detailed epidemiological data. In response, we have revised the introduction to incorporate additional information on demographic patterns and regional prevalence (see lines 62-65). Specifically, we note that males are more commonly affected than females, adolescents have a higher prevalence than younger children, and the highest rates are reported among children in or from South America and Asia. As this study focuses on exercise interventions, we have kept the epidemiological background concise while ensuring that the most relevant global context is included.

2. Has the renaming of NAFLD to MASLD resulted in any modifications to diagnostic criteria or clinical guidelines? If affirmative, what was the impact on your research?

Response: The renaming from NAFLD to MASLD reflects a conceptual shift rather than changes in the core diagnosis. Pediatric guidelines have not made any modifications yet. To maintain consistency, we used the term "NAFLD" throughout the systematic review since most included studies were published before the nomenclature change. We have now added text clarifying that the terminology change did not impact the eligibility criteria of the included studies (see lines 145–148).

3. The topics addressed in the final paragraph of the introduction pertain to methodology and discussion sections.

Response: We appreciate the reviewer’s comment and have revised the final paragraph of the introduction (see lines 87-94).

4. The inclusion of children with a BMI at or above the 85th percentile is rational but necessitates further examination of its implications. What was the author's rationale for including children with a BMI ≥ 85th percentile, and how does this impact the study's findings? What are the differences in outcomes between overweight and obese children?

Response: The inclusion of children with a BMI ≥ 85th percentile is justified as it aligns with clinical guidelines for screening MASLD. Our aim was to capture the full spectrum of children at risk, including those with overweight or obesity, to ensure comprehensive coverage. We acknowledge that the original methods section did not fully explain this rationale, and we have now revised it to provide further clarification (see lines 138-141). However, the included studies did not provide sufficient information to determine differences in response between overweight and obese children, as most were not designed or powered to do so.

5. Elucidate the rationale for incorporating non-randomized and uncontrolled studies, considering their intrinsic bias risk, as this affects the overall quality and conclusions of the review.

Response: We agree that non-randomized and uncontrolled studies carry a higher risk of bias. However, given the limited availability of RCTs specifically focused on pediatric exercise interventions for MASLD, we included these studies to provide a broader perspective. To mitigate their effect on the overall quality of the review, we grouped studies based on study design in the narrative synthesis (exercise vs. non-exercise control, uncontrolled studies of exercise intervention, exercise plus lifestyle intervention vs. lifestyle intervention alone, exercise vs. another type of exercise) and in Table 2 (RCT, NRCT, uncontrolled studies). Their intrinsic bias risk is also reflected in the risk of bias assessments (see Fig 3). Additionally, we have included a discussion of the potential impact of including these studies on the review's conclusions in the "Strengths and limitations" section (see lines 894–896).

6. What was the author's rationale for selecting participants aged 19 years or younger? What criteria did you use to define this age range as children and adolescents?

Response: We followed the World Health Organization's classification, which defines children and adolescents as individuals up to 19 years of age. This framework aligns with prior studies in pediatric hepatology and is reflected in our inclusion criteria. In the United States, pediatric gastroenterology clinics often continue to manage patients into early adulthood, including those aged 18 and 19. Thus, it was reasonable to include studies that enrolled participants within this age range. As outlined in the study populations, the ages of participants correspond to those commonly seen in clinical practice. We have added an explanation of this rationale in the methods section “Types of participants” (see lines 141-145).

7. Provide more details on how “higher risk for NAFLD” is defined. Were there any additional risk factors considered beyond overweight and obesity?

Response: Thank you for highlighting the need for greater specificity regarding our use of the term “higher risk for NAFLD.” Clinical guidelines recommend overweight and obesity as primary criteria for identifying children at increased risk for NAFLD and as the basis for determining which children should be screened. Accordingly, our review focused on studies involving populations with a confirmed diagnosis of NAFLD or where NAFLD was a target for intervention due to BMI ≥ 85th percentile. We have clarified this phrasing in the manuscript (see lines 526-527 and 664-665).

8. Explain how the diversity in intervention types (e.g., aerobic vs. resistance training) was managed in the synthesis. Were any subgroup analyses planned or performed?

Response: We acknowledge the heterogeneity in exercise interventions. We conducted a narrative synthesis rather than a meta-analysis due to this heterogeneity. Additionally, no subgroup analyses were planned or performed due to the limited number of studies available on this topic. This decision is now specified in the methods section (see lines 399-400). We have also commented on the diversity in intervention types in the discussion section (see lines 763-764).

9. Provide a rationale for selecting the three outcome categories: hepatic steatosis, liver chemistries, and liver histology. Were there any additional outcomes considered and excluded? If so, why?

Response: We selected hepatic steatosis, liver chemistries, and liver histology as outcome categories because they are the most clinically relevant and quantifiable indicators of liver health in MASLD. As explained in the manuscript, our focus was on quantitative measures to ensure objective assessment and comparability across studies. Conventional ultrasound was excluded as an outcome measure due to its subjective nature and inability to accurately quantify liver fat or detect subtle changes in response to therapy. This decision aligns with recommendations in the literature and ensures that the studies included in our review provide reliable, quantitative data. We have added this clarification to the methods section, along with relevant citations (see lines 173–258). 

10. Discuss any limitations of the search strategy, such as the exclusion of non-English studies or grey literature. Were any efforts made to identify unpublished studies?

Response: Our extensive search process did identify unpublished studies, including conference abstracts. However, our inclusion criteria required that studies be published in peer-reviewed literature. Including unpublished studies that have not undergone peer review or lacked the detailed reporting typical of full manuscripts would have introduced a high risk of bias. We have addressed the limitations of our search strategy, including this consideration, in the "Strengths and limitations" section of the discussion (see lines 892-894).

11. Elucidate the methods employed to rectify discrepancies. Were any particular criteria or thresholds employed?

Response: We followed a structured and systematic approach to address discrepancies, ensuring consistency and minimizing bias throughout the review process. Clear predefined eligibility criteria were applied to guide study selection and data extraction, with decisions informed by the protocols outlined in the manuscript. Thresholds for key assessments, such as study quality and risk of bias, were uniformly applied based on Cochrane guidelines. In cases where information was unclear or incomplete, studies were excluded only after consensus was reached among multiple reviewers. These methods reflect best practices in systematic reviews and were designed to enhance the transparency, rigor, and reliability of our findings.

Discrepancies in the selection of studies and data extraction were resolved through discussion between two review authors (MS and EY). If consensus could not be reached, a third review author (JS) was consulted to help make the final decision. This process is described in the methods section (see lines 299-301 and 359-360).

For the risk of bias assessments, a third review author (GM) conducted an independent evaluation of studies where disagreements arose. This additional review ensured that assessments were accurate and aligned with the Cochrane Risk of Bias 2 tool for randomized trials or the ROBINS-I tool for non-randomized studies. Final decisions were reached through further discussion among the full author team to ensure consistency across all assessments. This process is now described in more detail in the manuscript (see lines 409-411).

12. Discuss the implications of finding a high risk of bias in 10 out of 16 studies. How does this impact the overall conclusions and recommendations of the review?

Response: We conducted a rigorous analysis, including a thorough assessment of bias. Our findings reflect the current state of the field, making it essential to present these biases transparently. We acknowledge that the high risk of bias in 10 of the 16 studies limits the strength of our conclusions. This concern is now emphasized in the limitations section, along with recommendations for future research to address these gaps (see lines 897–899 and Fig 4).

13. Try to avoid bullet points as much as possible.

Response: We have removed all bullet points and presented the material in paragraph form (see lines 150-164, 172-173, 263-266, 362-375).

14. The discussion indicates that merely seven of the 16 studies included populations with a mean hepatic steatosis of ≥ 5%, and only six exhibited elevated baseline ALT levels. This indicates a limited scope that restricts the generalizability of results. Articulate the justification for the inclusion of studies featuring less severe hepatic conditions. How does this inclusion affect the interpretation of exercise's impact on more severe cases of NAFLD/MASLD?

Response: We included studies designed to treat NAFLD/MASLD, even though many involved participants with mild disease or populations where only a subset had the condition. This reflects a persistent challenge in the field that has not been adequately addressed, as studies continue to be designed without accounting for this limitation. We deliberately highlighted this issue to provide transparency and to compare the characteristics of participants in exercise studies with those in other clinical trials for children with MASLD. In this manuscript, we emphasize the need for future studies that specifically evaluate the impact of exercise on more severe cases of MASLD.

15. The discussion recognizes that negligible alterations in ALT and GGT were noted, attributed to baseline values being nearly normal. Could there be confounding factors or issues with the sensitivity of measurements?

Response: ALT and GGT are widely used, highly standardized clinical laboratory assays, with minimal potential for confounding in their measurement. Their reliability and consistency across clinical settings make them valuable indicators of liver health.

16. Cross-check the data mentioned in the discussion with the results section to ensure consistency.

Response: We appreciate your recommendation. We cross-checked the data in the discussion with the results section and have provided further clarification to ensure consistency in our presentation of the results (see lines 758-762).

17. The discussion indicates that numerous studies depend on MRI hepatic fat fraction and liver chemistry. This prompts apprehensions regarding the uniformity and comparability of results. Discuss the implications of using different assessment tools on the comparability of study results. How might these differences affect the overall conclusions drawn from the systematic review?

Response: The use of different outcome measures across studies reflects the current reality in both clinical research and patient care. Our goal was to include outcomes that are quantitative, clinically relevant, and widely accepted to provide the most comprehensive picture possible. We also addressed the variability in these measures and explained why this heterogeneity precluded conducting a meta-analysis. Throughout the review, we ensured a careful and transparent presentation of individual study results and synthesized them thoughtfully into the broader conclusions of the systematic review.

18. The segment regarding future directions is significant yet deficient in specificity. Offer more comprehensive and precise suggestions for subsequent research endeavors. Which specific study designs, populations, or methodologies should be emphasized?

Response: Thank you for the suggestion. We have expanded and provided more specific recommendations for future research directions (see Fig 4).

---

## [Editor Report · Decision Letter 2]

13 Nov 2024

Systematic review of exercise for the treatment of pediatric metabolic dysfunction-associated steatotic liver disease

PONE-D-24-08084R2

Dear Dr. Schwimmer,

We’re pleased to inform you that your manuscript has been judged scientifically suitable for publication and will be formally accepted for publication once it meets all outstanding technical requirements.

Kind regards,

Shaghayegh Khanmohammadi

Academic Editor

PLOS ONE
---

## [Editor Report · Acceptance letter]

29 Nov 2024

PONE-D-24-08084R2 

PLOS ONE

Dear Dr. Schwimmer, 

I'm pleased to inform you that your manuscript has been deemed suitable for publication in PLOS ONE. Congratulations! Your manuscript is now being handed over to our production team.

Kind regards, 

on behalf of

Dr. Shaghayegh Khanmohammadi 

Academic Editor

PLOS ONE